# Monitoring Stimulated Darkening from UV-C Light on Different Bean Genotypes by NMR Spectroscopy

**DOI:** 10.3390/molecules27072060

**Published:** 2022-03-23

**Authors:** Marília Vilela Salvador, Flávio Vinícius Crizóstomo Kock, Isabella Laporte Santos, Jean Fausto Carvalho Paulino, Caléo Panhoca de Almeida, Rodrigo Henrique dos Santos Garcia, Luciana Lasry Benchimol-Reis, Luiz Alberto Colnago, Antonio Gilberto Ferreira

**Affiliations:** 1Departamento de Química, Universidade Federal de São Carlos, Rodovia Washington Luís, São Carlos 13565-905, Brazil; mariliavilelasalvador@gmail.com (M.V.S.); giba@ufscar.br (A.G.F.); 2Centro de Recursos Genéticos Vegetais, Instituto Agronômico de Campinas, Av. Barão de Itapura 1481, Campinas 13075-630, Brazil; isa-laporte@hotmail.com (I.L.S.); jeanbiotec@gmail.com (J.F.C.P.); caleoalmeida@hotmail.com (C.P.d.A.); luciana.reis@sp.gov.br (L.L.B.-R.); 3Instituto de Química de São Carlos, Universidade de São Paulo, Avenida Trabalhador São Carlense 400, São Carlos 13566-590, Brazil; rodrigogarciaquimico@yahoo.com.br; 4Embrapa Instrumentação, Rua Quinze de Novembro 1452, São Carlo 13561-160, Brazil; luiz.colnago@embrapa.br

**Keywords:** *Phaseolus vulgaris* L., bean darkening, UV-C irradiation, NMR spectroscopy and chemometrics

## Abstract

The use of UV-C cool white light on bean (*Phaseolus vulgaris* L.) seeds significantly increases the biochemical seed coat post-harvest darkening process, whilst preserving seed germination. The aim of this work consists in monitoring the effect caused by the incidence of UV-C light on different bean genotypes using NMR spectroscopy. The genotype samples named IAC Alvorada; TAA Dama; BRS Estilo and BRS Pérola from the Agronomic Institute (IAC; Campinas; SP; Brazil) were evaluated. The following two methodologies were used: a prolonged darkening, in which the grain is placed in a room at a controlled temperature (298 K) and humidity for 90 days, simulating the supermarket shelf; an accelerated darkening, where the grains are exposed to UV-C light (254 nm) for 96 h. The experiments were performed using the following innovative time-domain (TD) NMR approaches: the RK-ROSE pulse sequence; one- and two-dimensional high resolution (HR) NMR experiments (^1^H; ^1^H-^1^H COSY and ^1^H-^13^C HSQC); chemometrics tools, such as PLS-DA and heat plots. The results suggest that the observed darkening occurs on the tegument after prolonged (90 days) and accelerated (96 h) conditions. In addition, the results indicate that phenylalanine is the relevant metabolite within this context, being able to participate in the chemical reactions accounted for by the darkening processes. Additionally, it is possible to confirm that a UV-C lamp accelerates oxidative enzymatic reactions and that the NMR methods used were a trustworthy approach to monitor and understand the darkening in bean seeds at metabolite level.

## 1. Introduction

The common bean (*Phaseolus vulgaris* L.) is a relevant crop that occupies a leading position in Brazilian agriculture. Brazil is the third largest producer, and the largest consumer, of this grain in the world [1]. However, beans commonly depreciate due to the post-harvest seed coat darkening phenomenon, which results in loss of their technological quality, causing their rejection by consumers. This is mainly due to dark-colored grains being misunderstood as being older and demanding more cooking time, which reduces their commercial value [2]. Therefore, studies aiming to improve the technological quality of beans, by monitoring the post-harvest seed coat darkening of these grains, may supply consistent keys to chemical and metabolic pathways, which are indispensable for the development of new elite cultivars in bean breeding programs.

Previous studies have shown that phenolic compounds present in the tegument can undergo chemical changes that lead to new compounds, which meaningfully contribute to the darkening process of the bean grains [3,4]. Post-harvest seed coat darkening is related to proanthocyanidin accumulation and its subsequent oxidation in the seed coat [5,6]. These compounds are found in higher concentrations in beans that are of normal darkening compared to those of slow darkening [7]. Grain darkening is due to several genetic, environmental, and post-harvest factors and storage conditions, such as humidity conditions, the drying time of the grains and, most importantly, the storage environment [4,8,9].

There are at least the following three types of phenotypic classes for the post-harvest seed coat darkening phenomenon of grains, as established by bean breeding programs: (1) non-darkening (ND), (2) slow darkening (SD) and (3) regular darkening (RD) [3,10,11]. In Brazil, among the several commercial kinds of available bean grains, the ‘carioca’ (70% of the market) and ‘black’ (15% of the market) varieties predominate [5]. Carioca beans have cream-colored grains with brown stripes [12,13], and belong to the Mesoamerican gene pool. The carioca commercial group is more prone to post-harvest seed coat darkening compared to black beans, despite some carioca bean cultivars having meaningful differences in their own darkening pattern.

Bean seeds are commonly studied and characterized using digital imaging processing [14], electrophoretic approaches [15], microbiological analysis [16,17], infrared spectroscopy, such as advanced vibrational molecular spectroscopy (ATR-FT/IRS) [18], near-infrared reflectance (NIR) [19,20,21], Fourier Transform (FT)-NIR [22,23], advanced chromatography methods, such as high-performance liquid chromatography, coupled with time-of-flight mass spectrometry [24] and spectrofluorimetry [25]. All these spectroscopies can supply the composition and quality evaluation for these seeds; however, they are not able to supply their metabolomic profile. Therefore, there is a need for an answer at molecular level regarding the post-harvest seed coat darkening phenomenon.

For this purpose, modern ^1^H-NMR spectroscopy methods are used, such as the time domain (TD) NMR approach named Rhim and Kessemeier-Radiofrequency Optimized Solid-Echo (RK-ROSE) sequence. These methods are able to provide an accurate estimation of rigid components in a quick, non-destructive way, and without the use of organic solvents [26]. In addition, molecular alterations on the metabolomic profile were evaluated, using high resolution (HR) NMR spectroscopy approaches, among which were ^1^H, ^1^H-^1^H COSY and ^1^H-^13^C HSQC. Furthermore, the NMR results were analyzed using metabolomic databases, such as FooDB [27], HMDB [28] and Chenomx [29]. Finally, discriminant chemometric studies, such as partial least-squares analysis (PLS-DA), were addressed so as to statically discriminate between the bean seeds in function of the metabolic alterations that occurred as a result of UV-C ray incidence.

Therefore, this work aimed to monitor and understand, at molecular level, the post-harvest seed coat darkening for the following bean cultivars: ‘IAC Alvorada’ (ND), ‘TAA Dama’ (SD), ‘BRS Style’ (SD) and ‘BRS Pérola’ (RD). The results may offer a deeper insight into the phenomena and indirectly help future development of new cultivars with better technological features and higher economic value.

## 2. Experimental

### 2.1. Prolonged and Accelerated Darkening Study

The cultivars ‘IAC Alvorada’, ‘TAA Dama’, ‘BRS Estilo’ and ‘BRS Pérola’ were selected due to the contrast observed in their post-harvest seed coat darkening. The experiment was performed in the experimental station named ‘Fazenda Santa Eliza’ (Instituto Agronômico, IAC, Campinas, Brazil), in September 2018. The harvest took place in December 2018 and was carried out manually. The pods of each cultivar were harvested individually and stored at room temperature for 10 days to homogenize the color. After this time, a sample of seeds sufficient to complete a 6 cm petri dish (approximately 50 grains) was separated for the darkening studies. Color readings were always performed in a closed room with fluorescent lighting using the colorimeter Minolta^®^ (model CR-410), in agreement with the international commission on illumination (CIE) 1976 system (L*a*b) standards [30], where L* represents the luminosity degree, ranging from black (0) to white (100), a* corresponds to the color scale between green (−60) and red (+60), and b* represents the color scale ranging from blue (−60) to yellow (+60), respectively.

Accelerated darkening was performed following the methodology proposed by Junk-Knievel et al. [31]. Each sample was placed in a petri dish (6 cm) without a lid and exposed to ultraviolet cool white light (UV-C lamps 254 nm) at full fluorescence for 96 h. The samples were positioned 10 cm below the lamps at an average intensity of 10.7 mW/cm^−2^ and inverted on the exposed side every 12 h. For each 24 h period, the seeds of every single sample were mixed. Shelf darkening evaluation was also performed, which aimed to simulate the natural darkening that occurs in a supermarket. For this purpose, each sample was packed in practical ziplock bags (12.5 cm × 8.5 cm) and these were kept on the shelf for 3 months (90 days) at temperature 294K and for a controlled photoperiod (12 h/12 h). The arrangement of the samples on the shelf and the side facing the light were randomized every 7 days, aiming at the homogenization of the samples.

### 2.2. NMR Analysis

#### 2.2.1. TD-NMR Analysis (CPMG and ROSE)

Relaxometry analyses were performed with a Bruker Mq-20 NMR spectrometer (20MHz for the ^1^H nucleus) at room temperature (298K), and using, for this purpose, 5 bean seeds from each genotype inside the NMR tube. These studies were performed using the traditional Carr-Purcell-Meiboom-Gill (CPMG) pulse sequence, besides the new solid-state sequence named RK-ROSE, which is able to refocus dipolar and quadrupolar interaction, allowing it to be used as a simple filter to rule out signals from mobile components in heterogeneous materials [26].

The main experimental parameters used for the CPMG sequence were an acquisition time of 0.02 min, time between echoes (δ) of 0.05 min, recycling delay (RD) of 1 s and 30,000 echoes. For the RK-ROSE experiments, 64 scans were acquired with an acquisition time of 3 min, focusing time of 1 s and using a power of 56 dB for the signal acquisition from the bean seeds.

#### 2.2.2. High-Resolution NMR

The samples were prepared in triplicates. For this purpose, 60.0 mg of the grain was crushed and solubilized in 800 µL of deuterated water (D_2_O) containing 0.002% (*w*/*v*) of TMSP-d_4_ (trimethylsilyl sodium propionate). The solution was vortexed for 5 min and then submitted to an ultrasonic bath for 5 min and centrifuged for 15 min at 14.500 rpm. Finally, 500 µL of the supernatant was transferred to the 5 mm NMR tube for the analysis.

All ^1^H NMR experiments were performed with a Bruker AVANCE III 9.4 Tesla spectrometer (400 MHz for ^1^H and 100 MHz for ^13^C nucleus, respectively) with a 5 mm inverse detection probe equipped with field gradient on the z axis, ATMA (Automatic Tuning Matching) and SampleXpress (automatic sampler).

The optimal results for the ^1^H-NMR measurements were reached using a pre-saturation pulse sequence with continuous wave (zgcppr-Bruker denomination) for the pre-saturation of the solvent residual water signal (HDO). In total, 64 scans (ns) were performed with 64k points during the acquisition (td) in a spectral window (sw) of 20 ppm and with a receiver gain (rg) equal to 512. The following parameters were used for acquisition: 3.0 s for relaxation delay (d1), 4.1 s for acquisition time (aq), 44.97 dB for power attenuation of pre-saturation, and of processing: with 64k points (SI), using exponential multiplication (lb) 0.3 Hz and automatic phase and baseline correction. The processing for the ^1^H-NMR data was performed automatically in each spectrum, adjusting the phase, baseline, and calibration by the internal reference (TMSP-d_4_) signal at 0.0 ppm, using for this purpose the software TopSpin [32] version 3.5 pl.7 from Bruker©.

### 2.3. Chemometric Analysis

#### 2.3.1. Multivariate Statistical Analysis

Multivariate analysis was applied to a total of 36 spectra from the 4 addressed bean genotypes. The data matrices were built after processing the NMR spectra, and a comparative analysis was performed throughout the darkening conditions (prolonged and accelerated). For this purpose, the data were pre-processed using normalization and scaling to remove possible bias arising due to sample handling and sample variability. Normalization (by median) was conducted in order to minimize possible differences in concentration between the bean samples. Following normalization, scaling (mean-centering and division by the square root of standard deviation of each variable) was performed to give all variables equal weight, regardless of their absolute value. After data pre-processing, Partial Least Squares Discriminant Analysis (PLS-DA) was performed using the web-based metabolomic data processing tool MetaboAnalyst 5.0 [33].

#### 2.3.2. Quantitative Analysis of Variation in Selected Metabolites

Metabolites meaningfully expressed (high area signal intensity) were chosen from the ^1^H-NMR spectra for quantitative measurements by spectral integration. It was assumed that any reduction in the signal intensity due to relaxation effects would be consistent across samples, and thus would not affect the evaluation of relative changes in metabolite levels. The peaks were integrated using TopSpin [32] software and the TMSP-d_4_ reference area signal was normalized to 1. Finally, the metabolites which showed clearer signals in the spectral window from δ 0–10 ppm were chosen, performing 20 variables, as can be observed in Appendix A.

## 3. Results and Discussion

A preliminary result about the variability of the composition of bean seeds caused by the incidence of the UV-C light was obtained by TD-NMR. Figure 1 shows the signals for the IAC Alvorada cultivar obtained from the use of CPMG (A) and RK-ROSE sequences (B), under conditions of prolonged darkening after 90 days of exposure to the fluorescent lamp with a 12-h photoperiod, alternating 12 h in the presence of light and 12 h in the dark, and accelerated darkening with exposure to a 254 nm wavelength UV-C lamp, which accelerates the biochemical processes of darkening, but preserves seed germination [3,31]. The results obtained for other bean genotypes can be observed in Appendix A.

Interestingly, the results do not show meaningful variations specifically between the genotypes. However, it is clearly possible to observe a slight decrease in the relaxation profiles during the darkening process (as a function of incidence of UV-C light time). The results demonstrate a decrease of the transverse (T_2_) relaxation times assessed by CPMG pulse sequence. This decrease, shown in Figure 1a, suggests an increase in the molecular restrictions, i.e., a possible polymer aggregation on the bean surfaces, as the result of an increase in the UV-C incidence time, from initial conditions (red curve) to final conditions, achieved after 96 h under UV-C incidence (blue curve), respectively. These observations were experimentally confirmed using the RK-ROSE pulse sequence in Figure 1b, which is able to provide an accurate estimation of rigid components in a fast, straight-through, and non-destructive way [26]. From this, using the TD-NMR approach, it is possible to confirm an increase in the mobile phase present in the intact bean seeds, consistent with changes in the tegument coloring. This facilitates a consistent initial picture about the effect of UV-C time incidence on bean seeds during the darkening process.

Furthermore, to investigate in more detail, at the molecular level, the darkening processes on the bean seeds, ^1^H NMR metabolomic studies were performed. Therefore, as can be observed in Figure 2, from the representative ^1^H-NMR spectra, it was possible to preliminarily identify the major classes of metabolites present in the TAA Dama cultivar. It is interesting to note that highlighted metabolites can be observed over all the ^1^H-NMR spectral window. Furthermore, a complete list of metabolites identified by 2D-NMR approaches (COSY, ^1^H-^13^C-HSQC and ^1^H-^13^C-HMBC) and using metabolomic databases, Chenomx [29], FooDB [27] and HMDB^28^, can be observed in Appendix A.

To discriminate the variables (metabolites) that have the greatest impact on grain composition under the different darkening conditions addressed, classification models using partial least squares analysis (PLS-DA) were applied using the MetaboAnalyst 5.0 version program [34]. Figure 3 shows the PLS-DA results for the initial samples from all genotypes (red highlight) and the two darkening groups after 90 days (green highlight) and after exposure to 96 h of UV-C light (blue highlight). These exploratory results showed a good separation for the two different darkening conditions, but were not able to discriminate between the bean genotypes.

Therefore, to thoroughly investigate, at metabolite level, the darkness phenomenon in the addressed beans seeds, a list of metabolites, according to the Variable Importance Projection (VIP), was used (Figure 4a). For this purpose, compounds with VIP higher than 1 are considered statistically significant, and therefore have a higher metabolic influence in the darkening of the bean seeds. The analysis of these variables showed that phenylalanine was the most significant metabolite able to separate the initial group of grains under different darkening conditions. This result points out that this amino acid is crucial in the study of grain darkening, suggesting a close relation, in consonance with literature, to procyanidin (otherwise known as condensed tannin) [35], a metabolite responsible for the darkening of the seed coat. Procyanidins are oligomers or polymers of flavan-3-ols (e.g., catechin and epicatechin), derived from the flavonoid biosynthesis pathway (Figure 4b) [5,7,35,36].

Furthermore, it is interesting to note in the heatmap plot (Figure 5) that the initial time and the grain exposed to 96 UV-C are metabolically similar. These results suggest that the accelerated darkening method occurs mainly in the tegument (external part) and not significantly in the cotyledon (internal part). These conclusions are supported by the results obtained by Junk-Knievel et al. [31], which described that the exposure of seeds to long darkening periods showed a low germination percentage, in comparison to seed exposure to short darkening periods, which presented a germination percentage similar to non-darkened seeds. As reported previously in darkening of cranberry seeds [31], germination was delayed and consistently lower over the non-darkening genotypes and this different profile was related to procyanidin content within the seed coats of these two kinds of genotypes (darkening and non-darkening). We speculate that, in the experimental conditions addressed in this work, the seed coat darkening not only darkens the tegument layer but also ages the cotyledons and seed embryos while, on the other hand, accelerated darkening only darkens the teguments of the grains, but does not age the cotyledon or seed embryos.

In addition, in Figure 5 it is possible to observe that darkened beans present important sugar metabolites, among them, raffinose, alpha and beta sucrose, xylose and amino acids, such as valine, isoleucine and leucine, after 90 days’ exposure. On the other hand, during accelerated darkening, the following important metabolites presented: stachyose sugars, verbascose and amino acids, among them being threonine, asparagine, and valine. Thus, it is possible to conclude that light incidence and time were not limiting factors to decrease the amounts of sugars, which are a crucial carbon source for germination [31,37] and energy metabolism in plants [38]. Therefore, it is possible to conclude that the conditions of long and short darkening incidence times, under the same conditions of temperature and humidity, do not appear to be a crucial factor to regulate bean germination, but just accelerate the darkening of the grain.

## 4. Conclusions

Given the results, it is possible to conclude that the observed darkening process occurs on the tegument (external part), under prolonged (90 days) and accelerated (96 h) conditions, and does not significantly occur in the cotyledon (internal part). In addition, the results point out that phenylalanine is the relevant metabolite within this context, suggesting a close relation with procyanidin, a relevant metabolite responsible for darkening on several grains in agricultural sciences. Furthermore, it was observed that the UV-C light incidence and time, in the conditions addressed in this work, were not limiting factors to decrease the concentration of sugars, which are essential to germination. In summary, this work demonstrates great potential to understand the post-harvest seed coat darkening phenomenon, at the metabolite level, while also paving the way for further research regarding the study of other economically relevant legumes that present similar post-harvest challenges, such as peas and soybeans.

## Figures and Tables

**Figure 1 molecules-27-02060-f001:**
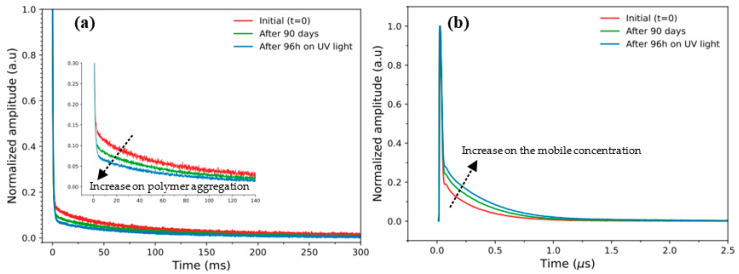
TD-NMR results obtained from (**a**) CPMG and (**b**) RK-ROSE signals for the ‘IAC Alvorada’ cultivar at zero time (initial color) (T0, red line), prolonged darkening after 90 days exposure to fluorescent light (12-h photoperiod) (T90, green line) and accelerated darkening after 96 h of exposure to UV-C light (T96, blue line).

**Figure 2 molecules-27-02060-f002:**
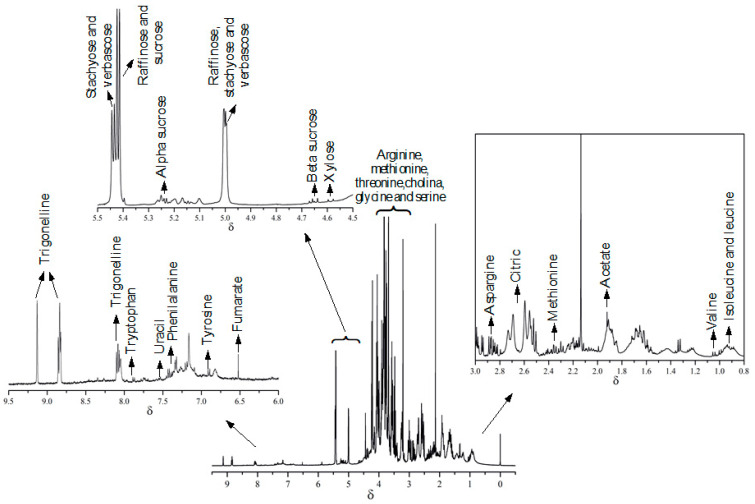
^1^H-NMR spectra obtained at 9.4 T (400 MHz for 1H) under initial conditions (*t* = 0) for TAA Dama cultivar.

**Figure 3 molecules-27-02060-f003:**
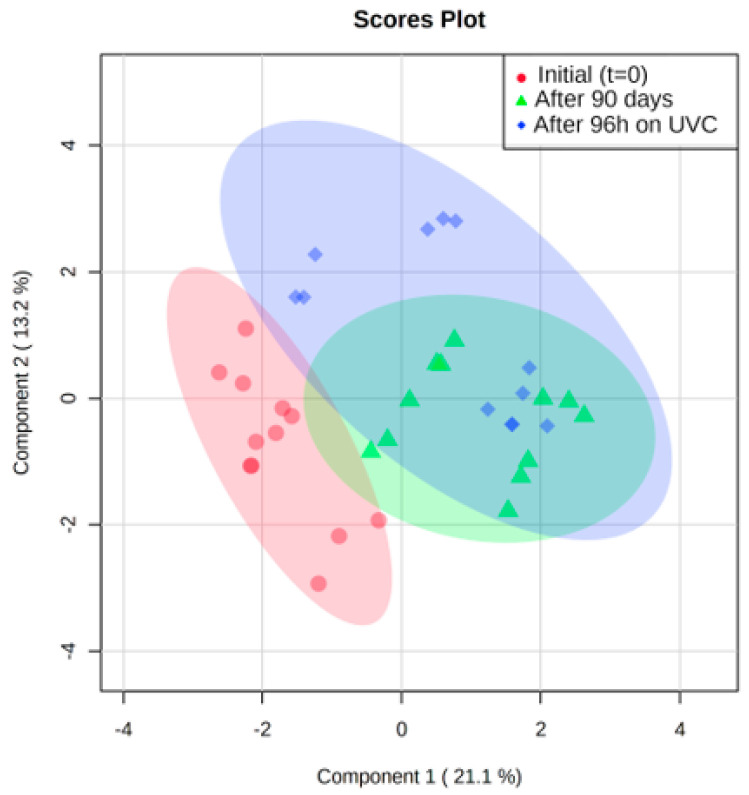
PLS-DA results to initial samples (*t* = 0), after 90 days and after 96 h on UVC of all genotypes: ‘IAC Alvorada’, ‘TAA Dama’, ‘BRS Estilo’ and ‘BRS Pérola’.

**Figure 4 molecules-27-02060-f004:**
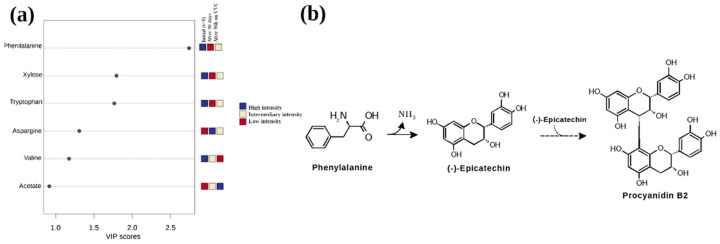
(**a**) Relevant metabolites discriminated by PLS-DA in a VIP projection. The colored boxes on the right indicate the relative area intensity of the corresponding metabolites in each of the groups studied. (**b**) Simplified reaction of the proposed model of the procyanidin biosynthesis pathway in the tegument from phenylalanine, in agreement with observations in this work for the bean genotypes addressed.

**Figure 5 molecules-27-02060-f005:**
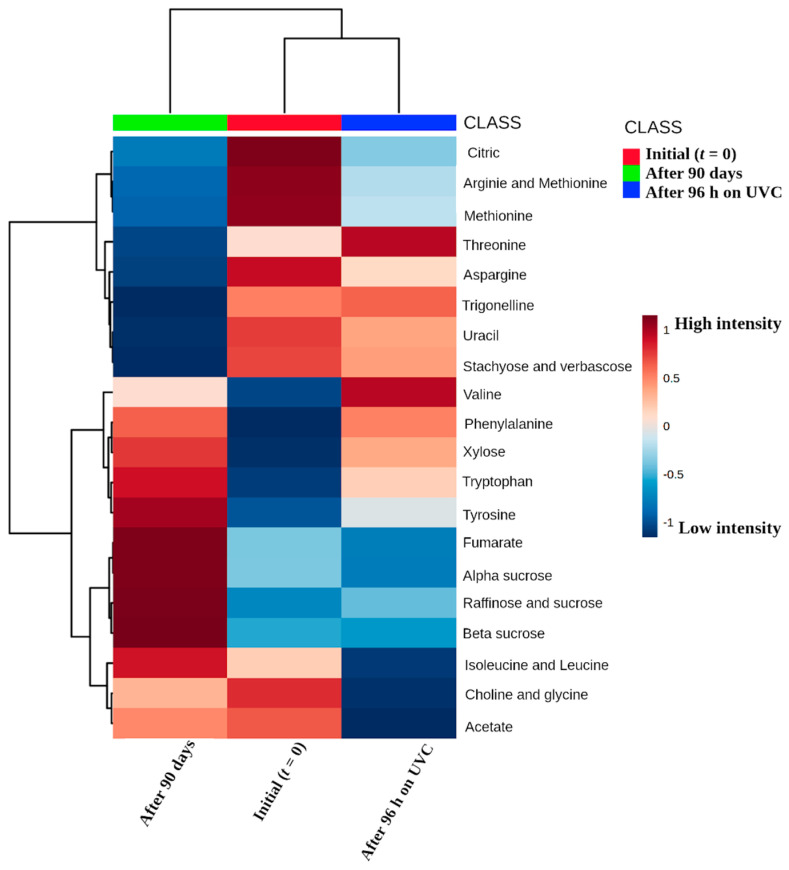
Visualization to the heatmap plot representing chemical composition similarity relationships with the different conditions of the darkening. The dark red color represents the maximum intensity, while the dark blue color represents the lowest intensity, respectively.

## Data Availability

Publicly available datasets are available from authors under request.

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
