# Peer review of "Monitoring Stimulated Darkening from UV-C Light on Different Bean Genotypes by NMR Spectroscopy"

_molecules, 2022, doi:10.3390/molecules27072060_

Round 1

Reviewer 1 Report

Major remarks:

The subject of the presented study is interesting and methods used are advanced, but do not the Authors think that the observed effects related to irradiation with the mercury line of 254 nm at quite high power, could not be caused by an increase in temperature in seeds, because the temperature was not stabilized? Whether they used a water filter behind the lamp to cut-off thermal radiation?  On the other hand, stabilizing the temperature in such kind of an experiment is certainly not easy.
Can the authors better describe the observed bound and unbound water phase changes by discussing the TD-NMR results?

Generally, Chapter 3 should be better described.

There are also some minor mistakes some of which are listed below:

Figure 4A is unreadable, no colors or error bars are visible in the chart.

Figure 5 must be much better describe in figure and in the text, now is unclear.

Author Response

The authors would like to thank the reviewer suggestions and comments that surely improved the final version of this work. The detailed answers for all considerations done by the reviewer are thoroughly discussed point by point, in the attached file. 

Thanks loads, 

Flavio Kock

Reviewer 2 Report

The manuscript “Monitoring the stimulated darkening from UV-C light on different beans genotypes by NMR spectroscopy” demonstrates the darkening of beans when subjected to UV-C light and natural shelf conditions. The authors use the NMR technique and chemometric methods for data processing. In general, the manuscript is clear, concise and presents satisfactory results. Minor language corrections need to be made.

In lines 60-65 – I suggest mentioning, in addition to the other instrumental methods, the application of digital images in common beans quality control. https://doi.org/10.1016/j.foodchem.2021.130349

PLS-DA is a discrimination tool, consider revisiting the text. Some authors understand the word “classify and its derivations” as an error in this context.

In the future, I encourage authors to use exploratory analysis methods, such as Principal component analysis. Due to the low number of samples/spectra, supervised methods such as PLS-DA may show false results.

In lines 222 the authors mention the PLS-DA cannot classify the genotypes. If the authors created the PLS-DA model only to discriminate the different experimental methods, it is not possible to make this statement. A new model considering genotypes as classes should be carried out, but I believe it is unnecessary in this manuscript.

Author Response

(The authors gave the same response as above.)
